# SoftTarget Regularization

## An effective technique to reduce over-fitting in Neural Networks

**Armen Aghajanyan**
Dimensional Mechanics
Bellevue, WA 98007, USA
`armen.aghajanyan@dimensionalmechanics.com`

## Abstract

Deep neural networks are learning models with a very high capacity and therefore prone to over-fitting. Many regularization techniques such as Dropout, DropConnect, and weight decay all attempt to solve the problem of over-fitting by reducing the capacity of their respective models (Srivastava et al., 2014), (Wan et al., 2013), (Krogh & Hertz, 1992). In this paper we introduce a new form of regularization that guides the learning problem in a way that reduces over-fitting without sacrificing the capacity of the model. The mistakes that models make in early stages of training carry information about the learning problem. By adjusting the labels of the current epoch of training through a weighted average of the real labels, and an exponential average of the past soft-targets we achieved a regularization scheme as powerful as Dropout without necessarily reducing the capacity of the model, and simplified the complexity of the learning problem. SoftTarget regularization proved to be an effective tool in various neural network architectures.

## 1 Introduction

Many regularization techniques have been created to rectify the problem of over-fitting in deep neural networks, but the majority of these methods reduce models capacities to force them to learn general enough features. For example, Dropout reduces the amount of learn-able parameters by randomly dropping activations, and DropConnect extends this idea by randomly dropping weights (Srivastava et al., 2014), (Wan et al., 2013). Weight decay regularization reduces the capacity of the model, not by dropping learn-able parameters, but by reducing the space of viable solutions (Krogh & Hertz, 1992).

### 1.1 Motivation

Hinton has shown that soft-labels, or labels predicted from a model contain more information that binary hard labels due to the fact that they encode similarity measures between the classes (Hinton et al., 2015). Incorrect labels tagged by the model describe co-label similarities, and these similarities should be evident in future stages of learning, even if the effect is diminished. For example, imagine training a deep neural net on a classification dataset of various dog breeds. In the initial few stages of learning the model will not accurately distinguish between similar dog-breeds such as a Belgian Shepherd versus a German Shepherd. This same effect, although not so exaggerated, should appear in later stages of training. If, given an image of a German Shepherd, the model predicts the class German Shepherd with a high-accuracy, the next highest predicted dog should still be a Belgian Shepherd, or a similar looking dog. Over-fitting starts to occur when the majority of these co-label effects begin to disappear. By forcing the model to contain these effects in the later stages of training, we reduced the amount of over-fitting.

## 1.2 METHOD

Consider the standard supervised learning problem. Given a dataset containing inputs and outputs, $X$ and $Y$, a regularization function $R$ and a model prediction function $F$ we attempted to minimize the loss function $\mathcal{L}$ given by:

$$\mathcal{L}(X, Y) = \frac{1}{N} \sum_{i=0}^{N} \mathcal{L}_i(\mathcal{F}(X_i, \mathbf{W}), Y_i) + \lambda R(\mathbf{W}) \tag{1}$$

where $\mathbf{W}$ are the weights in $\mathcal{F}$ that are adjusted to minimize the loss function, and $\lambda$ controls the effect of the regularization function. For our method to fit into the supervised learning scheme we altered the optimization problem by adding a time dimension $(t)$ to the loss function:

$$\mathcal{L}^t(X, Y) = \frac{1}{N} \sum_{i=0}^{N} \mathcal{L}_i^t(\mathcal{F}(X_i, \mathbf{W}), Y_i) + \lambda R(\mathbf{W}) \tag{2}$$

SoftTarget regularization requires into two steps: first, we kept an exponential moving average of past labels $\hat{Y}^t$, and second, we updated the current epochs label $Y_c^t$ through a weighted average of the exponential moving average of past labels and of the true hard labels:

$$\hat{Y}^t = \beta \hat{Y}^{t-1} + (1 - \beta)\mathcal{F}(X_i, \mathbf{W}) \tag{3}$$

$$Y_c^t = \gamma \hat{Y}^t + (1 - \gamma)Y \tag{4}$$

Here, $\gamma$ and $\beta$ are hyper-parameters that can be tuned to specific applications. The loss function then becomes:

$$\mathcal{L}^t(X, Y) = \frac{1}{N} \sum_{i=0}^{N} \mathcal{L}_i^t(\mathcal{F}(X_i, \mathbf{W}), Y_c^t) + \lambda R(\mathbf{W}) \tag{5}$$

The algorithm also contains a 'burn-in' period, where no SoftTarget regularization is done and the model is trained freely in order to learn the basic co-label similarities. We will denote the number of epochs trained freely as $n_b$, and the total number of epochs as $n$. Experimentally we also discovered that it is sometimes best to run the network for more than one epoch on a single $Y_c$, so we will denote $n_t$ as the number of epochs per every time-step. We have provided the pseudo-code in Algorithm 1.

---

**Algorithm 1** SoftTarget Regularization

**input**: $X, Y, \mathcal{F}, \mathcal{G}, \beta, \gamma, n_b, n_t, n$

$\mathcal{F} \leftarrow \mathcal{G}(\mathcal{F}, \{X, Y\}, n_b)$

$\hat{Y}^0 \leftarrow \mathcal{F}(X_i, \mathbf{W}), t \leftarrow 1$

**for** $i \leftarrow 0$ **to** $\frac{n - n_b}{n_t}$ **do**

 $\hat{Y}^t = \beta \hat{Y}^{t-1} + (1 - \beta)\mathcal{F}(X_i, \mathbf{W})$

 $Y_c^t = \gamma \hat{Y}^t + (1 - \gamma)Y$

 $\mathcal{F} \leftarrow \mathcal{G}(\mathcal{F}, \{X, Y_c^t\}, n_t), t \leftarrow t + 1$

**end**

---

Here $\mathcal{G}$ represents the training of the neural network, taking in a model $\mathcal{F}$, dataset $\{X, Y\}$ and an integer representing number of epochs.

A large $n_t$ allows the network to learn a better mapping to the intermediate soft-labels and therefore allows the regularization to be more effective. But increasing $n_t$ has a diminishing effect, because

as $n_t$ becomes large the network begins to over-fit to those soft-labels, and reduces the effect of the regularization, as well as increasing the training time of the network significantly. $n_t$ should be optimized experimentally through standard hyper-parameter optimization practices. We found $n_t = \{1, 2\}$ to work best through standard grid hyper-parameter optimization (Bergstra & Bengio, 2012). The small $n_t$ insures that the model does not overfit to the intermediate representation introduced by SoftTarget.

Through hyper-parameter optimization the same range of $\{1, 2\}$ was found to be optimal in the experiments we ran for $n_b$. A small $n_b$ insures that the co-label similarities captured by SoftTarget would not have been affected by any type of overfitting. This insures that as the experiments are further ran the true co-label similarties are propagated correctly. More complicated learning scenarias where the amount of labels and data is greater, the chances of corruption in co-label similarties is reduced and therefore larger $n_b$ can be choosen.

## 1.3 Similarities to Other Methods

Other methods similar to this are specific to the case where the $\beta$ hyper-parameter is set to zero, with no burn-in period.

- Reed et al. study the specific case of the SoftTarget method described above with the $\beta$ parameter set to zero (Reed et al., 2014). They focus on the capability of the network to be robust to noise, rather than the regularization abilities of the method.

- Grandvalet and Bengio have proposed minimum entropy regularization in the setting of semi-supervised learning (Grandvalet & Bengio, 2005). This algorithm changes the categorical cross-entropy loss to force the network to make predictions with high degrees of confidence on the unlabeled portion of the dataset. Assuming cross-entropy loss with SoftTarget normalization with a zero burn-in period, and zero $\beta$, our algorithm becomes equivalent to a softmax regression with minimum entropy regularization.

- Another similar approach to minimum entropy regularization is an approach called pseudo-labeling. Pseudo-labeling tags unlabeled data with the class predicted highest by a learning model (Lee, 2013). No soft-targets are kept, instead the predicted label is binarized, i.e. the highest class is labeled with a value of one, and every other class is labeled with a value of zero. These hard pseudo-labels are then fed as input to the model.

- Hinton et al described the power of soft targets in the use of transferring knowledge from one model to another, usually to a model that contains less parameters (Hinton et al., 2015). Soft-Target regularization can be interpreted as weighted distillation where the donor model is the state of the model at some previous time in training, and the weighting target are the hard-targets.

## 2 Experiments

We conducted experiments in python using the Theano and Keras libraries (The Theano Development Team, 2016), (Chollet, 2015). All of our code ran on a single Nvidia Titan X GPU, while using the CnMEM and cuDNN (5.103) extensions, and we visualized our results using matplotlib (Hunter, 2007). We used the same seed in all our calculations to insure the starting weights were equivalent in every set of experiments. The only source of randomness stemmed from the non-deterministic behavior of the cuDNN libraries.

## 2.1 MNIST

We first considered the famous MNIST dataset (LeCun et al., 1998). For each of the experiments discussed below, we performed a random grid-search over the hyper-parameters of the optimization algorithm, and a very small brute force grid search was done for the hyper-parameters of SoftTarget regularization. We compared our results to the cases where the hyper-parameters resulted in the best performance of the vanilla neural network without SoftTarget regularization. All of our reported values were computed on the standardized test portion of the MNIST dataset, as provided by the Keras library. The networks were trained strictly on the training portion of the dataset. We tested on

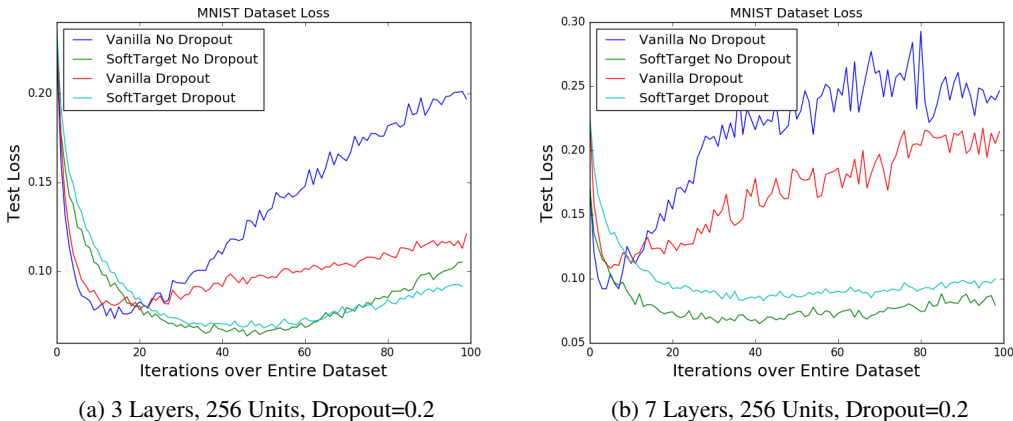

(a) 3 Layers, 256 Units, Dropout=0.2 (b) 7 Layers, 256 Units, Dropout=0.2

Figure 1: Regularization applied to multilayer neural networks.

eight different architectures, with four combinations of every architecture. The four combinations stem from testing each architecture via a combination of: no regularization, Dropout, SoftTarget, and Dropout+SoftTarget regularization.

We used a fully connected network, with a varying amount of hidden layers, and a set constant of neurons throughout each layer. Dropout was not introduced at the input layer, but was introduced at every layer after that. All of the layers activations we're rectified linear units (ReLu), except for the final layer which was a SoftMax. The net was trained using a categorical cross-entropy loss, and the ADADELTA optimization method. (Zeiler, 2012).

The frozen hyper-parameters for the SoftTarget regularization were: $n_b = 2, n_t = 2, n = 100, \beta = 0.70, \gamma = 0.5$. Our results are described in Table 1. We described the nets using the notation: $4 \leftarrow 256$ denoting a 4 hidden layer neural network, with each of the hidden layers having 256 units. We reported the minimum loss during training, the loss at the 100th epoch, and the maximum accuracy reached respectively.

Table 1: MNIST Comparison: minimum loss, loss at 100th epoch, max accuracy

| Net | Vanilla | SoftTarget | SoftTarget+Dropout (0.2) | SoftTarget+Dropout (0.5) | Dropout (0.2) | Dropout (0.5) |
|---|---|---|---|---|---|---|
| $4 \leftarrow 256$ | 0.076\|0.208\|0.981 | 0.063\|0.095\|0.982 | 0.068\|0.102\|0.989 | 0.114\|0.143\|0.974 | 0.081\|0.150\|0.983 | 0.137\|0.198\|0.978 |
| $5 \leftarrow 512$ | 0.077\|0.206\|0.984 | 0.056\|0.069\|0.986 | 0.060\|0.113\|0.985 | 0.101\|0.117\|0.978 | 0.087\|0.164\|0.984 | 0.088\|0.170\|0.976 |
| $6 \leftarrow 256$ | 0.199\|0.334\|0.979 | 0.063\|0.092\|0.990 | 0.075\|0.101\|0.982 | 0.148\|0.150\|0.985 | 0.101\|0.202\|0.981 | 0.086\|0.252\|0.970 |
| $6 \leftarrow 512$ | 0.079\|0.241\|0.981 | 0.056\|0.068\|0.990 | 0.064\|0.131\|0.985 | 0.131\|0.159\|0.977 | 0.089\|0.190\|0.981 | 0.152\|0.339\|0.978 |
| $7 \leftarrow 256$ | 0.092\|0.246\|0.981 | 0.065\|0.079\|0.985 | 0.083\|0.100\|0.983 | 0.207\|0.222\|0.978 | 0.108\|0.215\|0.977 | 0.216\|0.232\|0.968 |
| $7 \leftarrow 512$ | 0.090\|0.244\|0.982 | 0.056\|0.077\|0.985 | 0.071\|0.107\|0.985 | 0.172\|0.211\|0.978 | 0.099\|0.236\|0.983 | 0.175\|0.383\|0.974 |
| $3 \leftarrow 256$ | 0.074\|0.197\|0.981 | 0.064\|0.105\|0.985 | 0.068\|0.092\|0.990 | 0.109\|0.145\|0.975 | 0.079\|0.121\|0.985 | 0.118\|0.155\|0.980 |
| $3 \leftarrow 1024$ | 0.065\|0.138\|0.982 | 0.055\|0.088\|0.983 | 0.054\|0.084\|0.990 | 0.072\|0.112\|0.982 | 0.065\|0.138\|0.985 | 0.088\|0.137\|0.983 |
| $3 \leftarrow 2048$ | 0.065\|0.139\|0.982 | 0.053\|0.104\|0.983 | 0.052\|0.072\|0.990 | 0.060\|0.096\|0.990 | 0.071\|0.141\|0.978 | 0.088\|0.104\|0.987 |

In all our experiments, the best performing regularization for all of the architectures described above included SoftTarget regularization. Two representative results are plotted in Figure 1 for a shallow (three layer) and deep (seven layer) neural network. We saw that for deep neural networks (greater than three layers) SoftTarget regularization outperformed all the other regularization schemes. For shallow (three layer) neural networks SoftTarget+Dropout was the optimal scheme.

## 2.2 CIFAR-10

We then considered the CIFAR-10 dataset (Krizhevsky & Hinton, 2009), comparing various combinations of SoftTarget, Dropout and BatchNormalization (BN) (Ioffe & Szegedy, 2015). Batch-Normalization has been shown to have a regularization effect on neural networks due to the noise inherent to the mini-batch statistics. We ran each configuration of the network through sixty iterations through the whole training set. The complete architecture used was:

Table 2: CIFAR-10 Comparison

| Amount of Dropout | BN | SoftTarget | Just Dropout | SoftTarget+BN |
|---|---|---|---|---|
| 0 | 0.731\|1.876\|0.821 | 0.511\|0.592\|0.838 | 0.595\|1.120\|0.831 | 0.502\|0.540\|0.840 |
| 0.2 | 0.517\|0.855\|0.866 | 0.450\|0.501\|0.854 | 0.518\|0.706\|0.848 | 0.408\|0.410\|0.872 |
| 0.4 | 0.452\|0.596\|0.865 | 0.434\|0.478\|0.865 | 0.463\|0.543\|0.851 | 0.403\|0.432\|0.871 |
| 0.6 | 0.487\|0.560\|0.855 | 0.474\|0.488\|0.845 | 0.480\|0.550\|0.835 | 0.489\|0.526\|0.870 |
| 0.8 | 0.677\|0.741\|0.772 | 0.672\|0.695\|0.774 | 0.620\|0.714\|0.737 | 0.721\|0.777\|0.756 |

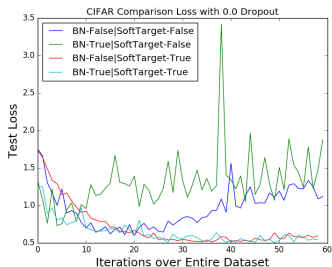

(a) No Dropout

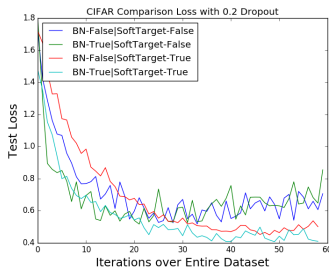

(b) Dropout=0.2

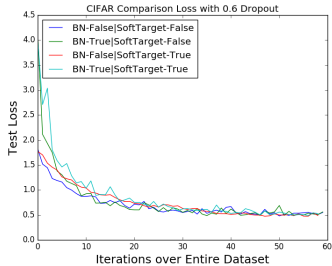

(c) Dropout=0.6

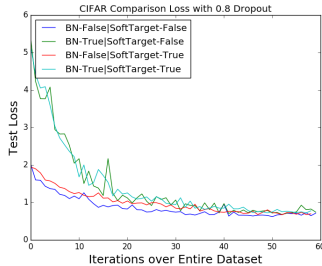

(d) Dropout=0.8

Input $\rightarrow$ Convolution (64,3,3) $\rightarrow$ BN $\rightarrow$ ReLU $\rightarrow$ Convolution (64,3,3) $\rightarrow$ BN $\rightarrow$ ReLU $\rightarrow$ Max-Pooling ((3,3), (2,2)) $\rightarrow$ Dropout ($p$) $\rightarrow$ Convolution (128,3,3) $\rightarrow$ BN $\rightarrow$ ReLU $\rightarrow$ Convolution (128,3,3) $\rightarrow$ BN $\rightarrow$ ReLU $\rightarrow$ MaxPooling ((3,3), (2,2)) $\rightarrow$ Dropout ($p$) $\rightarrow$ Convolution (256,3,3) $\rightarrow$ BN $\rightarrow$ ReLU $\rightarrow$ Convolution (256,1,1) $\rightarrow$ BN $\rightarrow$ ReLU $\rightarrow$ Convolution (256,1,1) $\rightarrow$ BN $\rightarrow$ ReLU $\rightarrow$ Dropout ($p$) $\rightarrow$ AveragePooling ((6,6)) $\rightarrow$ Flatten () $\rightarrow$ Dense (256) $\rightarrow$ BN $\rightarrow$ ReLU $\rightarrow$ Dense (256) $\rightarrow$ BN $\rightarrow$ ReLU $\rightarrow$ Dense (256) $\rightarrow$ SoftMax.

where: Convolution (64,3,3) signifies the convolution operator with 64 filters, and a kernel size of 3 by 3, MaxPooling ((3,3), (2,2)) represents the max-pooling operation with a kernel size of 3 by 3, and a stride of 2 by 2, AveragePooling ((6,6)) represents the average pooling operator with a kernel size of 6 by 6, Flatten represents a flattening of the tensor into a matrix, and Dense (256) a fully-connected layer (Krizhevsky et al., 2012), (Scherer et al., 2010). In our results, when we note that BN or Dropout weren't used, we simply omitted those layers from the architecture. We trained the networks using ADADELTA on the cross-entropy loss, using the same SoftTarget hyper-parameters we reported for the MNIST dataset. Our results are summarized in Table 2. The first column specifies the amount of Dropout used on the combinations listed in the next columns. As with the MNIST experiments, we reported the minimum loss during training, and the loss at the 100th epoch.

The use of SoftTarget regularization resulted in the lowest loss in four out of the five experiments on this architecture, and resulted in the lowest last epoch loss value and highest accuracy in all five of the experiments. As the dropout rate is increased the need for any other type of regularization is decreased. However, by increasing the rate of dropout, the resulting loss is increased because of the

reduced capacity of the network. SoftTarget regularization allowed a lower dropout rate to be used, and this lowered the test error.

## 2.3 SVHN

Finally, we considered the Street View House Numbers (SVHN) dataset, consisting of various images mapping to one of ten digits (Netzer et al., 2011). This is similar to the MNIST dataset, but is much more organic in nature, as these images contain much more natural noise, such as lighting conditions and camera orientation. We tested residual networks in four configurations: No regularization, Batch Normalization (BN), SoftTarget, and BN+SoftTarget (**?**). Our architecture consisted of the same building blocks as the residual network outlined by He et al., consisting of identity and convolution blocks (He et al., 2015). Identity blocks are blocks that do not contain a convolution layer at the shortcut, while convolution blocks do. In our notation I (3,[16,16,32], BN) will mean an identity block with an intermediate square convolution kernel size of 3, with three convolution blocks of size 16, 16 and 32. The outer convolutions contain kernel sizes of 1. C (3,[16,16,32], BN) contains the same initial architecture as I (3,[16,16,32]) but an additional convolution layer of size 32 at the shortcut connection. All of these blocks contained the rectified linear function as their activation, and BN prior to activation. Our final architecture was:

Input $\rightarrow$ ZeroPadding (3,3) $\rightarrow$ Convolution (64,7,7,subsample = (2,2)) $\rightarrow$ BN $\rightarrow$ ReLU $\rightarrow$ MaxPooling ((3,3), (2,2)) $\rightarrow$ C (3,[16,16,32], BN) $\rightarrow$ I (3,[16,16,32], BN) $\rightarrow$ I (3,[16,16,32], BN) C (3,[32,32,64], BN) $\rightarrow$ I (3,[32,32,64], BN) $\rightarrow$ I (3,[32,32,64], BN) $\rightarrow$ AveragePooling ((7,7)) $\rightarrow$ Dense (10) $\rightarrow$ SoftMax

We used the ADADELTA optimization method with a random grid search for hyper-parameter optimization. All configurations of the networks were run for 60 iterations apart from the overfit configuration which was run for 100 iterations.

We reported our results in Table 3 and Figure 3, as before giving the minimum test loss and the test loss at the last epoch. SoftTarget regularized configurations (with and without BN) again scored the lowest test loss and highest accuracy, compared to Batch Normalization alone.

Table 3: Residual Networks on SVHN

|  | No Regularization | BN | SoftTarget | SoftTarget+BN |
|---|---|---|---|---|
| Test Loss | 0.254—0.347 | 0.298—0.404 | 0.244—0.244 | 0.237—0.249 |
| Test Accuracy | 0.929—0.923 | 0.921—0.915 | 0.931—0.931 | 0.932—0.929 |

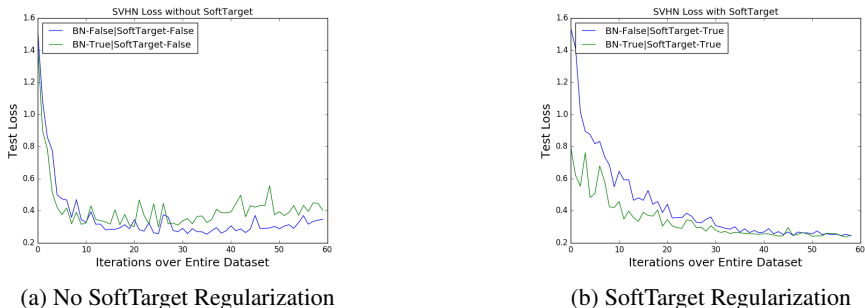

(a) No SoftTarget Regularization (b) SoftTarget Regularization

Figure 3: SoftTarget regularization applied to SVHN dataset.

## 2.4 CO-LABEL SIMILARITIES

We claimed that over-fitting begins to occur when co-label similarities that appeared in the initial stages of training, are not longer present. To test this hypothesis we compared the covariate matrices of a over-fitted network, early training stopped networks, and regularized networks. We tested again on the CIFAR10 dataset, with the same architecture as the previous CIFAR10 experiment,

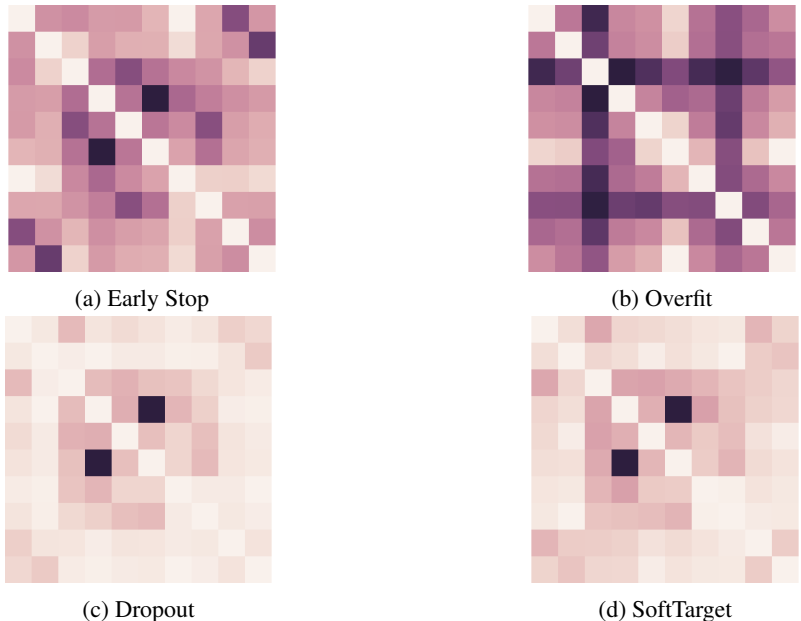

(a) Early Stop

(b) Overfit

(c) Dropout

(d) SoftTarget

Figure 4: Covariance matrices for the CIFAR10 dataset.

except that the number of filters and dense units were reduced exactly by two. We compared four configurations: Early (10 epochs), Overfit (100 epochs), Dropout ($p$=0.2, 100 epochs) and SoftTarget ($n_b = 2, n_t = 2, \beta = 0.7, \gamma = 0.5$, 100 epochs). After training each configuration for its respected amount we predicted the labels of the training set. We then calculated a covariance matrix scaled to a range of $[0, 1]$ since we are only interested in the relative co-label similarities. We set the diagonal to all zeros, as to make it easier to see other relations. The covariance function used is defined below.

$$c_{i,i} = 0 \tag{6}$$

$$c_{x,y} = \frac{\sum_{i=1}^{N}(x_i - \bar{x})(y_i - \bar{y})}{N - 1} \tag{7}$$

$$covs(x, y) = \frac{c_{x,y} - \min(c_{x,y})}{\max(c_{x,y}) - \min(c_{x,y})} \tag{8}$$

We plotted the covariance matrices in Figure 4. For the early stop case, there we observed the highest covariance between labels 3 and 5, which correspond to cats and dogs respectively. This intuitively makes sense, during earlier steps of training, the network learns to first detect differences between varying entities, such as frog and airplane, and then later learns to detect subtle difference. It is interesting to note, that this is the core principle behind prototype theory in human psychology (Osherson & Smith, 1981), (Duch, 1996), (Rosch, 1978). Some concepts are in nature closer to each other than others. Dog and cat are closer in relation than frog and airplane, and our regularization method mimics this phenomena. Another interesting thing to note is that the dropout method of regularization produces a covariance matrix that is very similar to that produced by SoftTarget regularization. The phenomena of co-label similarities being propagated throughout learning is not specific to just SoftTarget regularization, but regularization in general. Therefore co-label similarities can be seen as a measure of over-fitting.

## 3 CONCLUSION AND FUTURE WORK

In conclusion, we presented a new regularization method based on the observation that co-label similarities apparent in the beginning of training, disappear once a network begins to over-fit. SoftTarget

regularization reduced over-fitting as well as Dropout without adding complexity to the network, therefore reducing computational time, and we provided novel insights into the problem of over-fitting.

Future work will focus on methods to reduce the number of hyper-parameters introduced by Soft-Target regularization, as well as providing a formal mathematical framework to understand the phenomenon of co-label similarities.

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
