# Peer review of "SoftTarget Regularization: An Effective Technique to Reduce Over-Fitting in Neural Networks"

_ICLR 2017 — rejected_

[Reviewer Comment · AnonReviewer3 · 02 Dec 2016]
**Modeling and setup**

- You definitely need to report misclassification error results on test data for obvious reasons related to losses and final test misclassification error. Currently comparisons are not conclusive.

-  Can you explain better the reason for using the particular updates in (3) and (4) better? Why don't you do for example totally corrective update, e.g. take convex combination of all \cal{F}'s (or some portion) up to current iteration in (3)? Therefore \beta and \gamma should be tuned reasonably well to see whether (3) and (4) is really helping or not and the range for cross validation should be reported.

- The reason to set n_t n_b is not satisfactory.  It is crucial to cross-validate such parameters. Isn't  n_t = {1,2} unreasonably small number that can cause unstable results? why all n_b and n_t are equal?Are there results on other n_b and n_t's that were tried?

- It is stated that colabel similarities disappear when network starts to overfit. However distillation ( Hinton et.al. ,2015 ) captures colabel similarities after training a model and using distillation. This method seems an iterative extension of distillation without using a bigger teacher model. Does proposed method gives better results then a two step version of distillation ?  

- How do you tune \lambda for weight decay? 

- From paper: "We considered a frozen set of hyper-parameters for the SoftTarget regularization to show that SoftTarget regularization can still work without a having to conduct a large grid search". This argument is not valid in ML, maybe if you did a reasonable search, you would get worse results (since you should not look test error until you finish the cross-validation).   Why a common hyper parameter tuning procedure is not used e.g. random search (Bergstra and Bengio, JMLR 2012) or Bayesian optimization (Snoek et al ,NIPS 2012) ?  Setting the hyper parameters to some numbers without searching a range or set can dramatically ruin fair comparison.

[Official Review · AnonReviewer1 · rating 4 · confidence 5 · 15 Dec 2016]
**An interesting approach, but I'm unconvinced.**

This manuscript tries to tackle neural network regularization by blending the target distribution with predictions of the model itself. In this sense it is similar in spirit to scheduled sampling (Bengio et al) and SEARN (Daume et al) DAgger (Ross et al) which consider a "roll-in" mixture of the target and model distributions during training. It was clarified in the pre-review questions that these targets are generated on-line rather than from a lagged distribution, which I think makes the algorithm pseudocode somewhat misleading if I understand it correctly.

This is an incremental improvement on the idea of label softening/smoothing that has recently been revived, and so the novelty is not that high. The author points out that co-label similarity is better preserved by this method but it doesn't follow that this is causal re: regularization; a natural baseline would be a fixed, soft label distribution, as well as one where the softening/temperature of the label distribution is gradually reduced (as one would expect for this method to do as the model gets closer and closer to reproducing the target distribution).

It's an interesting and somewhat appealing idea but the case is not clearly made that this is all that useful. The dropout baselines for MNIST seem quite far from results already in the literature (Srivastava et al 2014 achieves 1.06% with a 3x1024 MLP with dropout and a simple max norm constraint; the dropout baselines here fail to break 1.3% which is rather high by contemporary standards on the permutation-invariant task), and results for CIFAR10 are quite far from the current state of the art, making it difficult to judge the contribution in light of other innovations. The largest benchmark considered is SVHN where the reported accuracies are quite bad indeed; SOTA for single net performance has been less than half the reported error rates for 3-4 years now. It's unclear what conclusions can be drawn about how this would help (or even hurt) in a better-tuned setting.

I have remaining reservations about data hygiene, namely reporting minimum test loss/maximum test accuracy rather than an unbiased method for model selection (minimum validation set error, for example). Relatedly, the regularization potential of early stopping on a validation set is not considered. See, e.g. the protocol in Goodfellow et al (2013).

[Official Review · AnonReviewer3 · rating 3 · confidence 5 · 19 Dec 2016 (modified: 23 Jan 2017)]
**the empirical results are not satisfactory**

Inspired by the analysis on the effect of the co-label similarity (Hinton et al., 2015), this paper proposes a soft-target regularization that iteratively trains the network using weighted average of the exponential moving average of past labels and hard labels as target argument of loss. They claim that this prevents the disappearing of co-label similarity after early training and  yields a competitive regularization to dropout without sacrificing network capacity.

In order to make a fair comparison to dropout,  the dropout should be tuned carefully. Showing that it performs better than dropout regularization for some particular values of dropout (Table 2) does not demonstrate a convincing advantage. It is possible that dropout performs better after a reasonable tuning with cross-validation.

The baseline architectures used in the experiments do not belong the recent state of art methods thus yielding significantly lower accuracy. It seems also that experiment setup does not involve any data augmentation, the results can also change with augmentation. It is not clear why number of epochs are set to a small number like 100 without putting some convergence tests.. Therefore the significance of the method is not convincingly demonstrated in empirical study.

Co-label similarities could be calculated using softmax results at final layer rather than using predicted labels.  The advantage over dropout is not clear in Figure 4, the dropout is set to 0.2 without any cross-validation.  


Regularizing by enforcing the training steps to keep co-label similarities is interesting idea but not very novel and the results are not significant.

Pros : 
- provides an investigation of regularization on co-label similarity during training

Cons:
-The empirical results do not support the intuitive claims regarding proposed procedure
Iterative version can be unstable in practice

[Official Review · AnonReviewer2 · rating 4 · confidence 5 · 20 Dec 2016]

The paper introduced a regularization scheme through soft-target that are produced by mixing between the true hard label and the current model prediction. Very similar method was proposed in Section 6 from (Hinton et al. 2016, Distilling the Knowledge in a Neural Network). 

Pros: 
+ Comprehensive analysis on the co-label similarity.

Cons:
- Weak baselines. I am not sure the authors have found the best hyper-parameters in their experiments. I just trained a 5 layer fully connected MNIST model with 512 hidden units without any regularizer and achieved 0.986 acc. using Adam and He initialization, where the paper reported 0.981 for such architecture. 
- The authors failed to bring the novel idea. It is very similar to (Hinton et al. 2016). This is probably not enough for ICLR.

[Final Decision · Program Chairs · 06 Feb 2017]
**ICLR committee final decision**

The reviewers unanimously recommend rejection.